# Advances in Spinal Cord Stimulation

**DOI:** 10.3390/bioengineering10020185

**Published:** 2023-02-01

**Authors:** Christopher M. Lam, Usman Latif, Andrew Sack, Susheel Govindan, Miles Sanderson, Dan T. Vu, Gabriella Smith, Dawood Sayed, Talal Khan

**Affiliations:** 1Department of Anesthesiology and Pain Medicine, University of Kansas Health System, Kansas City, KS 66160, USA; 2School of Medicine, University of Kansas, Kansas City, KS 66160, USA

**Keywords:** chronic pain, spinal cord stimulation, neuromodulation, spinal cord injury, artificial skin, electrodes

## Abstract

Neuromodulation, specifically spinal cord stimulation (SCS), has become a staple of chronic pain management for various conditions including failed back syndrome, chronic regional pain syndrome, refractory radiculopathy, and chronic post operative pain. Since its conceptualization, it has undergone several advances to increase safety and convenience for patients and implanting physicians. Current research and efforts are aimed towards novel programming modalities and modifications of existing hardware. Here we review the recent advances and future directions in spinal cord stimulation including a brief review of the history of SCS, SCS waveforms, new materials for SCS electrodes (including artificial skins, new materials, and injectable electrodes), closed loop systems, and neurorestorative devices.

## 1. Introduction

Chronic pain was systemically classified in the International Classification of Diseases 11 (ICD-11) by the International Association for the Study of Pain (IASP) working group in collaboration with the World Health Organization (WHO) in 2019. They defined chronic pain as pain that persists or recurs for more than 3 months with further subcategorization into six other subgroups including its own disease entity [1]. In the United States alone, it has been found that 50.2 million adults (roughly 20.5% of the US population) report pain daily with significant limitations to social activities and activities of daily living and with associated statistically significant increased days missed from work compared to people without chronic pain (10.3 v. 2.8); this includes an estimated gross domestic product impact of approximately $296 billion dollars lost in productivity, annually [2].

The predominant pain locations in those that responded to this 2019 National Health Interview Survey were back pain, hand/shoulder/arm pain, and hips/knee/foot pain [2]. With the paradigm shift away from predominantly pharmacologic management, historically reliant upon narcotic management due to concerns from the opioid epidemic, opioid sparing management techniques have become popularized including therapy, non-narcotic pharmacologic management, and interventions [3]. A staple of interventional management of chronic pain has been dorsal column spinal cord stimulation (SCS).

Initially utilized primarily for management of pain conditions including failed back syndrome, refractory angina pectoris, peripheral vascular disease, and complex regional pain syndrome (CRPS), SCS has become a mainstay in chronic pain management [4]. Within the past decade, new indications for SCS including management of painful diabetic neuropathy and non-surgical low back pain have increased their utilization in clinical practice [5,6]. Here we review the historical technologic nuances and recent advances in SCS devices (Figure 1).

## 2. History of Spinal Cord Stimulation

One of the earliest recorded forms of neuromodulation was by the ancient Romans in 15 A.D. to alleviate gout pain by incidental contact and subsequent electrical stimulation by torpedo fish (Figure 2) [7,8]. This early concept of transcutaneous electrical nerve stimulation (TENS) has evolved over time with the first modern machine for therapeutic electricity (“Electreat”) progressing to the modern day TENS unit pioneered by Dr. C. Norman Shealy [9]. Over time, advancements in the field of neuromodulation would result in the SCS systems that we have today.

The first commercially available SCS system was made by Medtronic in 1968, borrowing from their many developments and advances with cardiac pacemakers [8]. The following five decades would see advancements predominately in the hardware of these devices. Early systems consisted of an internal electrode that was surgically implanted via laminectomy. These internal electrodes were connected to a receiving antenna, which interfaced with an external generator and antenna via radiofrequency to generate the power needed for electrical stimulation [10]. By the 1980s, with the advent of lithium-based batteries, fully implantable systems were possible, with the implantable pulse generator (IPG) replacing the radiofrequency-based systems. Subsequently, percutaneously placed leads by needle placement was pioneered as an alternative means of lead placement without the need for a laminotomy [11].

The gate control theory of pain proposed by Malzack and Wall is the first proposed mechanism of action to explain SCS [12]. According to this theory, the transmission of pain signals depended on a balance of activity from larger A-beta fibers, smaller A-delta, and C fibers which modulated the transmission of pain by an inhibitory interneuron [12,13]. Dr. Norman Shealy also pioneered SCS by placing the first system in 1967. This initial configuration involved subdural electrodes placed at the dorsal columns at the level of T2-3. These electrodes were then connected to an external generator, where application of an electrical current 10 to 50 times a second relieved the patient’s chronic chest pain and abdominal pain [14]. It was believed that application of this electrical signal led to an activation of the larger A-beta fibers, leading to increased inhibitory interneuron activity and the closure of the “gate” to the transmission of pain signals. Further research has suggested that the mechanism for pain relief with conventional SCS is incompletely explained by the gate control theory; rather, there are numerous other spinal segmental and supraspinal mechanisms at play [15]. Current belief is that spinal cord stimulation differentially activates the dorsal horn of the dorsal column. Depending on the waveform and stimulation modality, differential activation of various rexed laminae, nerve cell types such as micro glial cells, or selective inhibitory pathway activation may account for the mechanism of action for this treatment modality [16,17,18].

The same timeframe also saw advancements in electrode technology used with SCS systems. Early SCS systems utilized electrodes that were surgically implanted subdurally via laminectomy; however, cerebrospinal fluid leak and spinal cord injury were potential complications from this approach. Another challenge for the successful use of SCS was the optimal selection of patients who would have long-term benefits from SCS. Both of these would provide the impetus for the development of electrodes that could be placed percutaneously via a needle. Early implementations placed percutaneous electrodes again in the subdural space, but quickly evolved to placement in the epidural space [19,20]. These percutaneous electrodes allowed for a trial of spinal cord stimulation in a much less invasive manner compared to a surgical laminectomy and were quickly incorporated for use in permanent implantations [21]. For these reasons, percutaneous electrodes became the predominant type utilized in SCS systems, and the older paddle electrodes were reserved for patients where percutaneous placement would be challenging. Percutaneous electrode placement also opened the door for SCS use by pain physicians, already familiar with epidural access, and are now responsible for the majority of SCS implants today [22].

The success of these early percutaneous systems was limited by therapy failure due to migration of the electrodes, which may necessitate a revisional procedure to reposition the electrode [21]. This limitation was mitigated, in part, by increasing the number of contacts on a single lead wire. Whereas early SCS systems had a lead with just a single contact, developments in lead design starting in the 1980s led to leads with an array of four contacts, which then increased exponentially to 8- and 16-contact arrays in percutaneous leads and 32-contact arrays in paddle leads. This, combined with advancements in IPG technology, allowed the programmatic configuration of anodes and cathodes along the electrode array to optimize stimulation at an individual level. Small lead migrations which would have necessitated revision before could now be compensated for by reselecting the electrodes programmatically. Further advancements in IPG processing allowed for each electrode in the array to be individually addressed and further modulate the stimulation delivered at each electrode, giving us the multichannel systems that are in use today [23,24]. All of these advancements in hardware would lay the framework for the advancements in spinal cord stimulation that would follow.

## 3. Waveforms

Expanding indications for the use of spinal cord stimulators, the increased acceptance by the medical community, and the aforementioned advancements in SCS hardware would all serve as catalysts for the rapid development of therapeutic programming. SCS waveforms are characterized by their frequency, duration (pulse width), amplitude, and pulse shape [15]. The prototypical waveform generated by SCS systems is known as conventional, or tonic, stimulation. Compared to the newer waveforms (including high-frequency, burst, and differential target multiplexed spinal cord stimulation), conventional stimulation is charactered by relatively lower frequency (40–80 Hz), higher pulse width (200–500 µS), and higher amplitude (3.5–8.5 mA), all of which result in a higher amount of charge per pulse delivered. This high charge per pulse leads to the generation of an action potential at the site of stimulation. Orthodromic activation of the large A-beta fibers results in the perception of paresthesias, a hallmark of conventional SCS. Numerous studies have suggested that paresthesias overlapping the area of pain is necessary for effective pain relief from conventional SCS [24,25]. This is achieved through the manipulation of the waveform amplitude and electrode placement via intraoperative mapping and post-operative programming to achieve a suitable paresthesia in the desired treatment area.

Given the duration of time that conventional tonic stimulation has been used for, especially compared to newer waveforms, there are numerous studies detailing its efficacy and mechanisms of action. One of the main mechanisms is believed to be antidromic activation of A-beta fibers, leading to the activation of inhibitory interneurons in the dorsal horn to attenuate pain signaling. Orthodromic activation is believed to lead to activation of supraspinal pathways that further increase the attenuation of pain [15]. However, there is a subset of patients for whom conventional SCS is ineffective, or its effectiveness diminishes over time, leading to the needed development of novel stimulation strategies.

Within the past decade, the impetus on novel waveforms has led to the discovery of high-frequency, burst, and differential target multiplex waveforms. A shared characteristic between these novel waveforms is that the stimulation is generally paresthesia-free as the efficacy of the waveform is not reliant on the generation of paresthesias. The rapid pace at which these novel waveforms are implemented clinically has outpaced the understanding on a basic science level; the elucidation of the mechanisms of action of these waveforms remains a topic of ongoing research.

High-frequency stimulation is characterized by high frequency (10 kHz), low pulse width (30 µS), and low amplitude (1–5 mA), which results in subthreshold stimulation that does not result in the generation of paresthesias. The lack of paresthesias with this waveform obviates the need for intraoperative paresthesia mapping; rather, the electrodes are placed anatomically based on the location of the pain. A randomized controlled trial comparing high-frequency stimulation to conventional stimulation demonstrated superiority for the treatment of both chronic back pain and chronic leg pain [26]. Recent studies aimed at determining the mechanism of paresthesia-free high-frequency stimulation suggest that 10 kHz SCS results in a selective activation of inhibitory interneurons in the spinal dorsal horn while sparing the activation of dorsal column fibers [27]. A study comparing rodent electrophysiology showed that 10 kHz but not burst signaling selectively activated the inhibitory interneurons in the dorsal horn without activating dorsal column fibers [16]. Further studies have explored the combination of high-frequency SCS with conventional SCS, with results suggesting a synergistic interaction between the two waveforms [28].

The principle of burst stimulation stems from the observation that the nervous system utilizes two patterns to transmit information, burst and tonic (as already seen with conventional SCS). Burst stimulation was designed to resemble this burst patterning, which is characterized by a 40 Hz burst frequency with a pulse width of 1000 µS that consists of spikes with an intraburst frequency of 500 Hz followed by a quiescent phase until the next burst cycle [29]. Several companies have touted burst stimulation in their programming portfolio. This wave form was developed by Dr. De Ridder. Dr. De Ridder noted that the difference between true burst patterning and other burst patterning mimics is that after the five monophasic spikes in true burst stimulation, there is a passive charge balance which imitates physiologic burst firing, whereas mimics utilize an active charge balance [30]. Multiple studies have concluded that spinal cord stimulation via a burst waveform is more efficacious compared to conventional stimulation [31,32]. While the current theory is that burst stimulation results in modulation of the behavioral and emotional responses to painful stimulus, further studies are needed to determine its mechanisms of action. Currently, the understanding is burst stimulation utilizes differential modulation of dorsal horn and dorsal column nuclei resulting in differential activation within the brain to result in pain control with implications of lateral and affective pathway modulation as well [17].

While the previous waveforms focused on the stimulation of neuronal cells, differential target multiplexed spinal cord stimulation (DTM-SCS) targeted the supportive glial cells in the nervous system. This paradigm was based on multiple observations: the predominance of glial cells at the site of SCS, the role of activated glial cells in chronic pain states, and that previously existing SCS strategies led to changes in gene expression of neural cells. The waveform for DTM-SCS is characterized by multiplexed signals with frequencies between 50 Hz and 1200 Hz with pulse widths between 50 and 400 µS [33,34,35]. A randomized controlled trial comparing DTM-SCS with conventional SCS demonstrated superiority of DTM-SCS over conventional SCS for chronic low back pain but failed to do so for chronic leg pain [36].

Though effective in managing pain, despite the variety of SCS waveforms available, the long-term success rate for SCS ranges from 47–74% [37]. There is no definitive single best waveform for various pain conditions, though new indications are emerging for SCS including diabetic neuropathy [5]. The exact mechanism of action for the various waveforms of SCS remains elusive and further research is needed.

## 4. Artificial Skins as Electrodes

Advancements in clinical medicine, informatics, and engineering have allowed for the emergence of wearable sensors and artificial skins. These skins or sensors may be able to monitor physiological changes that occur daily. Current applications for these range from the monitoring of clinical illness to the wellness and fitness arena [38]. Of interest is the utilization of these materials for use as spinal cord stimulator electrodes. There are a number of physical properties that skin is able to sense: pressure, strain, temperature, and light [39].

These artificial smart skins also have application in sensing bioelectrical signals, such as electromyography (EMG), electroencephalography (EEG), and electrocardiography (ECG). EEG/EMG is conventionally a rigid electrode combined with an electrolyte gel. Recent advancements by one group were able to show that electrodes can work as single conductive gels. This stretchable hydrogel was created by converting graphene oxide into conductive graphene through a reduction reaction. The performance of this hydrogel was comparable to that of commercially available electrodes [40].

Despite these advances, creating commercially viable artificial skin products presents a difficult challenge for material researchers. Currently in development are ultra-thin polymer foils that can conform to ambient and dynamic environments. These are organic transistors that contain an ultra-dense oxide gate dielectric that is a few nanometers thick and capable of repeated bending and stretching [41].

The fabrication of stretchable circuits with integrated transistors will allow for the next generation of skin electronics. Recently, a group was able to fabricate these intrinsically stretchable electronic polymers and able to achieve a skin-like character [42]. However, due to low gas permeability, most electronic skin devices do not allow for the secretion of sweat from the skin. This can lead to maceration and overhydration inhibiting the barrier function of skin [38]. The creation of a conductive and highly gas-permeable nano-mesh structure by one group may have overcome this issue [43]. Because of the increasing sophistication of materials used for smart skins, large-scale manufacturing will bring on a new set of challenges for the dissemination of this technology [38].

## 5. Implantable Electrode Material

Implanted electrodes is a robust field of interest in chronic pain management. Utilization of stimulator technology depends on effective electrode-cellular interface. These materials should be able to deliver stimulation to the target area without exceeding a cutoff threshold that could cause damage to the surrounding tissues [44]. Additionally, following implant, the body attempts to isolate the implant from the tissue. There are no side effects noted directly from SCS electrodes, though reports of allergies to spinal cord stimulator materials including electrodes and electrode malfunction have been reported [45,46]. Future materials aim to limit the foreign body response while still providing a durable, sensitive electrode [47].

One of the most important factors that influences the efficacy of stimulator function is impedance [44]. This factor is influenced by the types of materials used, the tissue-cellular interface, and the surface area of the electrode [48]. Increasing the size of the electrode can then allow for less impedance; however, this is often not feasible since limited space in the spine requires smaller electrodes that are resistant to bending and flexing [44,49].

Most commonly, substrates such as platinum/iridium, platinum, titanium, and gold are used for the stimulation and recording of electrodes. However, compared to bare electrodes, the addition of a coating allows for improved resolution [44]. Iridium oxide has been applied as a coating with hopes to improve neural stimulation [50]. The layers of iridium oxide offer improved injection capacity and lower impedance when compared to bare electrodes [49,50]. Despite the improved performance, the electrode structure is susceptible to degradation which could form cracks and leak particles into nearby tissues.

Despite a field that is largely dominated by metals, conductive polymers may offer improved electrical properties and reduced mechanical mismatch between electrodes and tissues [51]. Additionally, they are capable of housing and releasing drugs and bioactive materials [52]. These conductive polymers themselves are not conductive but work through a mechanism of doping. This means that a charge is introduced to the polymer chain by a dopant while preserving a neutral system [49,52]. The surface area of conductive polymers, due to nano structuring, is significantly increased, which allows for decreased impedance and improved capacitance [53].

Poly(3,4-ethylenedioxythiophene) (PEDOT) has become a popular option as a conductive polymer as it has shown to be the most stable conductive material for implantable devices [54]. Stability is still heavily dependent on the dopant. Doping PEDOT with polystyrene sulfonate (PSS) allows for significantly reduced interface impedance when compared to bare metal Pt (107 MΩ μm^2^ vs. 3900 MΩ μm^2^ at 1 kHz) [55]. This is due to its high ionic conductivity and large electroactive surface area [56]. Another important feature of PEDOT/PSS as an electrode material is its volumetric capacitor ability, which is roughly 100 times higher than common metal electrodes [55]. Despite the observed success, there are significant concerns about the long-term stability of PEDOT/PSS [54].

Doping PEDOT with carbon nanotubes (CNT) has demonstrated improved electrochemical stability when compared to that of PEDOT/PSS [57]. Recent in vivo studies measured the stability of PEDOT doped with CNT versus PSS. Researchers measured the chronic recording performance of the two materials and demonstrated that, over a four month period, PEDOT/CNT coated electrodes had superior recording stability [54]. Despite the demonstrated stability, concerns about shedding loose CNTs have been proposed. Early in vivo trials demonstrated good biocompatibility as results yielded significantly less neuronal death and damage along with decreased inflammatory response [58].

Another emerging material that is being researched for neural interfacing electrodes is Boron-doped diamond (BDD). This material offers a high degree of biocompatibility demonstrated by in vivo studies that found thinner fibrous capsules and a milder degree of inflammation at 2 and 4 weeks when compared to titanium nitride [59]. Recording performance of this material was limited due to high impedances and low double-layer capacitances [60]. However, more recent studies offer a solution.

The use of a three-dimensional nanostructured surface which the diamond is grown on demonstrated significant neural recording and stimulation performance improvement. When compared to conventional BDD microelectrodes, 3D-nanostructured BDD allowed for the detection of whole embryonic hindbrain-spinal cord preparations. This was most significant in the observation of low amplitude (10–20μV) [61]. This technique improves the merits for BDD as a material for electrodes in future devices.

## 6. Injectable Electrodes

Implanted devices utilized for recording and stimulation can aid patients suffering from a variety of neurological conditions. However, given the complexity and cost associated with these devices, some companies have looked for alternative methods to improve access for patients. One company has developed an injectable electrode (Injectrode^®^). This electrode is a flowable pre-polymer, and upon injection, cures to form a compliant neural electrode in vivo [62]. Given the ability to conform as needed, it may offer the ability to be utilized as an electrode for a variety of neuroanatomical targets.

The Injectrode^®^ was initially prepared by mixing platinum curing silicone elastomer and silver metallic flakes [62]. This creates an electrode with a Young’s modulus less than 100 kPa. This offers an electrode that is much less stiff than the current neural interface wires. The reduction in mechanical mismatch may lead to reduced strain and stress on the device and surrounding anatomy [62,63]. This concept has been demonstrated with previous soft electrodes which have improved electrode acceptance and reduced inflammatory response [64].

More recently, one team attempted to compare afferent fiber recruitment utilizing the Injectrode^®^ and DRG stimulation. This was achieved by exposing the L6 and L7 DRG in 4 cats. The DRG was then stimulated by either the Injectrode^®^ or a stainless-steel electrode. The antidromic evoked compound action potentials (ECAP) in various nerves throughout the body were then recorded. The recruitment rates and charge-thresholds were then calculated. This study demonstrated that the Injectrode^®^ recorded similar ECAP thresholds to that of the stainless-steel wire across all primary afferent neurons. Thus, the authors concluded that the Injectrode^®^ can stimulate primary afferent neurons and may be applicable in the clinical setting [65]. Despite the significant success, further biocompatibility studies will need to be conducted. This is because the initial Injectrode^®^ samples utilized silver, which has known toxic effects [66].

## 7. Closed-Loop Systems

The advent of closed-loop spinal cord stimulation was yet another major paradigm shift in regard to SCS strategies. While previous stimulation strategies operated via a fixed-output, open-loop manner, closed-loop stimulation allowed for the incorporation of real-time feedback. The linear electrode array design used in SCS systems allows for the evaluation of electrically evoked compound action potentials. A therapeutic stimulus from one of the electrodes leads to the generation of an ECAP, which then propagates both orthodromically and antidromically. The remainder of the unused electrodes can be utilized to measure the ECAP in both directions [67]. The difference between the charge generated at an electrode and the charge that is delivered to the spinal cord is directly proportional to the distance between the electrode and the target site. With the epidurally placed electrodes, this distance is primarily determined by the thickness of the dura and the dorsal CSF layer [68]. It is well established that postural changes and physiological movements (e.g., respirations, heartbeats) can result in minute changes in this distance, which can have a significant change in charge delivered per Coulomb’s law.

ECAPs are measurements of electrical response from the measured tissues of interest. In SCS, ECAPs are recordings of the total voltage change in the extracellular matrix that surrounds the bundles of axons and are measures of neural activation during SCS [69,70,71]. The ECAP electrodes are positioned at an optimal distance away from stimulating SCS electrodes in order to measure full potentials while minimizing artifact. Of note, artifacts are high-amplitude patterns that are caused by the SCS stimulation. Proper distancing between the stimulation and ECAP measurement electrodes allows time to differentiate between the artifact and ECAP because the artifact will present earlier than the ECAP itself. Short pulse widths, filtering, and template correlation are additional methods of reducing the artifact and sharpening the ECAP signals [69].

Each ECAP has a notable triphasic shape: a voltage peak (P1), negative deflection (N1), and second positive voltage peak (P2). The amplitude is the measured voltage difference between N1 and P2 [67,69]. ECAP with amplitude settings can be used to optimize spinal cord stimulator relief in open- and closed-loop systems. For example, when measured in the clinical setting, an optimal ECAP range for correlating clinical pain relief can be targeted and manually adjusted for during postural changes. Moreover, in closed-loop stimulator systems, ECAP feedback during distance changes between leads and target tissue can be used to automatically adjust stimulation levels, allowing for therapeutic compensation without manual manipulation. For optimal ECAP ranges, systems have suggested using the lowest amplitude that produces a perceptible paresthesia as the lower limit, while using the patient’s highest tolerable stimulation amplitude as the upper limit [69]. Thus, with the measurement of the generated ECAP and comparison to a target ECAP, SCS systems can utilize this real-time feedback to adjust current throughput accordingly to maximize the therapeutic effect [72].

Several studies evaluating the efficacy of ECAP systems in management of chronic pain have been promising. The Avalon Study was a prospective, multicenter, single-arm project that analyzed the Evoke System from Saluda Medical, the first closed-loop SCS system [70]. The study showed that 24 months after implant, back pain was reduced by a mean of 77.3% with the closed-loop system. In terms of ECAP amplitude, the median of most-used amplitude was 22.5, 28.5, and 25.5 uV at 3, 12, and 24 months, respectively. In conjunction, the patients were within the therapeutic window a median of 96.7, 84.9, and 90.2% of the time during the study at 3, 12, and 24 months [70].

The EVOKE trial was a multicenter, prospective, randomized, double-blind, parallel-group clinical trial that compared an open-loop to closed-loop SCS using the EVOKE SCS system by Saluda Medical. Approximately 82.3% of patients in the closed-loop group had at least 50% reduction in pain, with no increase in baseline pharmacologic medicines, while 60.3% in the open-loop group met the criteria for primary outcome at 3 months. At 12 months, 83.1% of patients in the closed-loop group and 61% in the open-loop group met the criteria for primary outcome [72]. When addressing ECAP amplitudes and the optimal therapeutic window, closed-loop patients, out-of-clinic, spent 91.1% of time within the therapeutic window, compared to 59.5% in the open-loop group at 3 months. At 12 months, patients within the closed-loop group spent 95.2% of time within the therapeutic window, compared to 47.9% in the open-loop group [72].

The Evoke study shows the potential impact of ECAP when combined with closed-loop SCS for consistent adjustment to maintain a therapeutic window. This allows patients to have sustainable pain control during physical movement without manual adjustment. Further studies using ECAP can likely elucidate the components of spinal cord activation, leading to optimal pain control through appropriate stimulation parameters.

## 8. Neurorestorative Devices

Closed-loop spinal cord stimulation has restorative implications for patients with spinal cord injury (SCI). Worldwide, SCI affects 10.4 to 83 per 1 million persons per year [73]. Disability from SCI is highly variable, ranging from mild weakness to tetraplegia with dependence on mechanical ventilation. Lifetime healthcare and living expenses are as high as $5.1 million per patient depending on age and the severity of injury [74]. This does not include indirect costs, including over $77,000 per year in lost wages and other benefits [74]. A patient’s degree of functional disability correlates with several negative outcomes: poor quality of life, reduced longevity, and mortality [75,76,77]. The relationship between functional disability and morbidity drives interest in true functional restoration for SCI patients.

Traditional medical treatment for SCI focuses on assistive equipment and physical therapy to optimize remaining motor function [73]. These help patients adapt to their level of disability but are limited in their ability to restore function. Over the past decade, functional electrical stimulation (FES) has emerged as an additional treatment modality. In FES, electrical stimulation is applied locally to a nerve or muscle to augment a specific weak motor activity [78]. Several devices exist, ranging from transcutaneous to fully implanted. However, these devices have significant limitations. They require triggering by the patient, either using a switch (e.g., the heel touching the ground when taking a step) or by voluntary activation of a nearby muscle [78]. FES also uses fixed stimulation output in response to triggering, not an adaptive response specific to the intended movement [78]. Tonic dorsal column spinal cord stimulation has also been used to augment strength in patients with incomplete SCI, but also suffers from the inability to finely adjust the desired motor response [79].

In contrast to FES, epidural electrical stimulation (EES) describes stimulation through electrodes implanted in the epidural space inside the spinal canal or placed transcutaneously over the spine. Scientific evidence has shown that EES of lumbosacral spinal circuitry can generate tonic and rhythmic patterns of motor activity in animals as well as humans diagnosed with complete SCI [80,81,82,83]. Moreover, EES in humans with complete loss of motor function after SCI enabled restoration of flexion and extension of leg movements, standing, and stepping following several months of training [84,85].

In 2021, ONWARD Medical, Inc. (Eindhoven, The Netherlands) launched a clinical trial (“Up-LIFT”) to evaluate the safety and efficacy of closed-loop spinal cord stimulation in patients with spinal cord injury [86]. The system is noninvasive and uses two transcutaneous electrodes placed over the cervical spine. When a patient initiates an upper extremity movement, the device uses stimulation to augment the motor response. Input from proprioceptive neurons is used to modify the strength and muscle selectivity of the motor stimulation. Subjects in the stimulation group were compared to subjects who instead underwent standard therapy called Functional Task Practice. Although published results are not available as of the writing of this review, ONWARD provided a press release stating, “The Up-LIFT pivotal study… reached its primary effectiveness endpoint of improvement in upper extremity strength and function” [87].

Another clinical trial is underway to evaluate the use of fully implanted epidural electrodes to improve lower extremity muscle function in patients with spinal cord injury In it, a new type of epidural lead was created for this purpose, since existing spinal cord stimulator leads were designed to stimulate sensory pathways in the medial spinal cord, not the lateral nerve roots which allow more precise muscle stimulation. The new lead contained 16 electrodes divided into 3 columns with more lateral spacing than standard spinal cord stimulator paddle leads. During operative lead placement, motor testing was performed to create a map in which muscles responded to stimulation of each electrode. The lead was physically connected to an implanted pulse generator (IPG). Multiple adhesive monitoring electrodes were placed on the subject’s legs to measure muscle response, then communicate this data wirelessly to the IPG. The IPG modified the stimulation output to achieve the desired muscle selectivity and intensity. All subjects received standard rehabilitation treatment throughout the study period. As of the writing of this review, the results of this study are not published. However, the authors did publish a case series of the first three subjects [88]. These subjects were unable to walk at baseline, even with body weight support. On the first day of stimulation, each subject was able to walk using a body-weight support system. Within 6 months of device implant, each subject could walk without human assistance or body weight support.

## 9. Conclusions

Spinal cord stimulation has quickly become a staple in interventional pain management for treatment of various chronic pain conditions. Great advances have been made since its initial inception in 1967, leading to new technologies allowing for minimally invasive placement and internalized batteries. With increased mechanistic understanding and implementation of new wave forms, innovations in hardware and software continue to push the field forward. Application of this technology for treatment of new indications such as neurorestoration after spinal cord injury is likely possible within the next decade. Concurrently, development of new materials allows for better programming and functioning electrodes. Spinal cord stimulation has established itself in the algorithm for management of chronic pain due to the ease of placement and breadth of conditions treated. Relatively easy to implant and explant, this therapy has been successfully utilized within the past few decades. However, further studies are needed to understand the mechanisms of action to improve the utilization of this device. Though effective, this therapy is associated with a long-term failure rate of 30–50%. With many advancements on the horizon, the field of neuromodulation will continue to adapt to improve efficacy and utilization as new technologies become available.

## Figures and Tables

**Figure 1 bioengineering-10-00185-f001:**
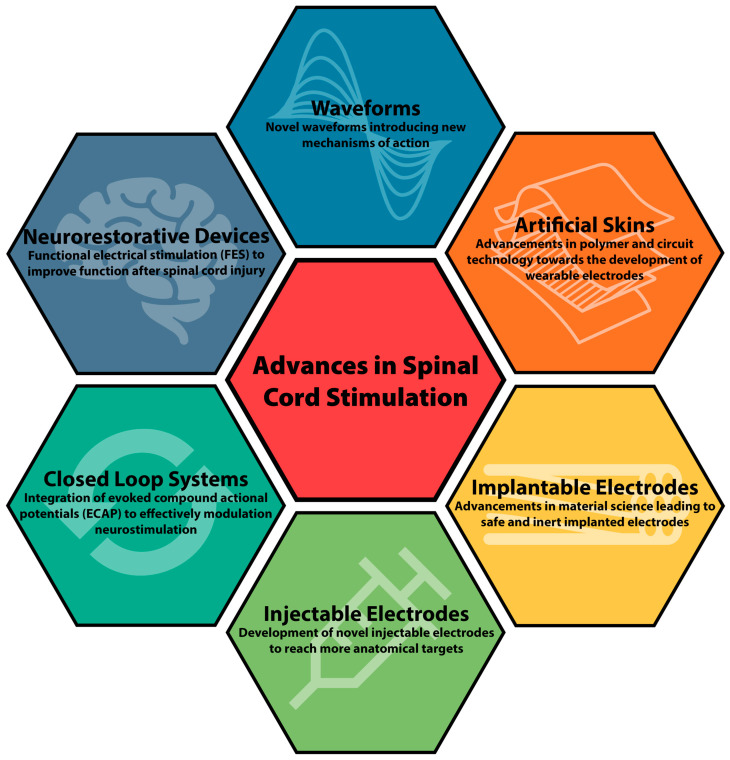
Schematic summary for advances in spinal cord stimulation.

**Figure 2 bioengineering-10-00185-f002:**
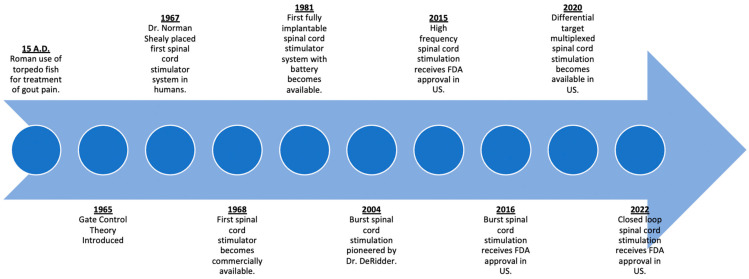
Timeline of advances in spinal cord stimulation.

## Data Availability

Not applicable. This is a review article and no original data were created by the authors in preparation of this manuscript.

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
