# Peer review of "Advances in Spinal Cord Stimulation"

_bioengineering, 2023, doi:10.3390/bioengineering10020185_

Round 1

Reviewer 1 Report

This paper reviews the recent advances and future directions in spinal cord stimulation. 

Strengths: The review is comprehensive and up-to-date.

Weakness: The review may be too long.

1. The reference [7] for electrical stimulation by torpedo fish differs somewhat from that listed on https://www.ncbi.nlm.nih.gov/pmc/articles/PMC7136297/

2. Please place the references for Malzack and Wall, and Norman Shealy at the end of the sentence where they are first mentioned (pg. 3).

3.  Please add newer waveform characteristics on pg. 4 line  119. Are these the same mentioned on pg. 5 line 44?

4. Please explain burst patterning (pg. 5, line 159). While several burst stimulation regimes have been proposed in literature, has there been a comparative study done to compare their efficacy?

5. Wondering what characteristic constitute for the lower impedance for PEDOT (pg. 7, line 238).

6. Please clarify the definition of ECAP and/or ECAP recordings on pg. 9, line 311.

7.  Wondering if the first two paragraphs of Section 8 would look better in the Introduction.

8. While most of this article makes an interesting reading, at times, lengthy procedures cited from a single reference make it somewhat boring (e.g., last paragraph on pg. 9 or the second and the third paragraphs on pg. 11).

Reviewer 2 Report

A review is interesting and sounds. It is a synthetic approach to the current state of knowledge about the importance of spinal cord stimulation in neuromodulation therapy. Before publication, it requires minor corrections, both editorial and clarifying the presented issues.

The abstract is "frugal". Authors should list in it all groups of issues presented in the text.

The authors reliably present the methodological aspects of spinal cord stimulation from a historical perspective, rarely criticizing the legitimacy of the methods used, especially various types of stimulation algorithms, often ineffective or even iatrogenic.

 Did all types of electrodes have no side effects?

The SCS mechanism of action has been presented. Any critical thoughts on the Melzack and Wall theory?  The sentence in lines 78-80 attempts to critically summarize the issue with only one ref. Have authors heard about the Hebbian learning theory? (Young W. (2015). Electrical stimulation and motor recovery. Cell transplantation, 24(3), 429–446. https://doi.org/10.3727/096368915X686904)

I think that the authors should treat the role of combined therapies with the use of SCS more broadly (upgrade of some refs. in this issue).

The STIMO study (lines 408-420)  is incomplete, not published in details, and should not be mentioned (it does not exist in PubMed).

The paper lacks critical SCS limitations after the long review. The advantages of the method should be summarized in the end.

 Part..."10. Patents

No patents held by the authors are resulting from the work reported in this manuscript. ... is redundant.

Although the authors put a lot of effort into writing the manuscript, it is not prepared according to the scheme favoured by Bioengineering. I think they should use a template and follow the guidelines specified there. For example, why are figure captions underlined?

References are listed in several styles, none of them MDPI style.

For example, 40. Journal of neural engineering. Small letters instead of big letters..., also in other refs.

46. Organic Conducting Polymers, in Organic Bionics. 2012. pp. 81-112. is misquoted

54. Trevathan, J.K., et al., An Injectable Neural Stimulation Electrode Made from an In-Body Curing 583

Polymer/Metal Composite. Adv Healthc Mater, 2019. 8(23): p. e1900892. change to..."Trevathan, J.K., Baumgart, I.W., Nicolai, E.N., Gosink, B.A., Asp, A.J., Settell, M.L., Polaconda, S.R., Malerick, K.D., Brodnick, S.K., Zeng, W., Knudsen, B.E. , McConico, A. L., Sanger, Z., Lee, J. H., Aho, J. M., Suminski, A. J., Ross, E. K., Lujan, J. L., Weber, D. J., Williams, J. C., ... Shoffstall, A. J. (2019). An Injectable Neural Stimulation Electrode Made from an In-Body Curing Polymer/Metal Composite. Advanced healthcare materials, 8(23), e1900892."...
